# Opposite effects of positive and negative symptoms on resting-state brain networks in schizophrenia

Xinrui Wang [1,2], Zhao Chang[1,2] & Rong Wang [1✉]

Schizophrenia is a severe psychotic disorder characterized by positive and negative symptoms, but their neural bases remain poorly understood. Here, we utilized a nested-spectral partition (NSP) approach to detect hierarchical modules in resting-state brain functional networks in schizophrenia patients and healthy controls, and we studied dynamic transitions of segregation and integration as well as their relationships with clinical symptoms. Schizophrenia brains showed a more stable integrating process and a more variable segregating process, thus maintaining higher segregation, especially in the limbic system. Hallucinations were associated with higher integration in attention systems, and avolition was related to a more variable segregating process in default-mode network (DMN) and control systems. In a machine-learning model, NSP-based features outperformed graph measures at predicting positive and negative symptoms. Multivariate analysis confirmed that positive and negative symptoms had opposite effects on dynamic segregation and integration of brain networks. Gene ontology analysis revealed that the effect of negative symptoms was related to autistic, aggressive and violent behavior; the effect of positive symptoms was associated with hyperammonemia and acidosis; and the interaction effect was correlated with abnormal motor function. Our findings could contribute to the development of more accurate diagnostic criteria for positive and negative symptoms in schizophrenia.

[1] College of Science, Xi'an University of Science and Technology, Xi'an, Shaanxi, China. [2] These authors contributed equally: Xinrui Wang, Zhao Chang. ✉email: wang0712_xust@163.com

Schizophrenia is a complex and severe psychotic disorder featuring impaired functions across multiple dimensions, including cognition, language, movement, emotion, and social behavior[1]. This disorder affects approximately 1% of the global population and results in considerable burdens on patients, families, and society[2,3]. In the clinic, schizophrenia is ordinarily diagnosed through the observation of positive symptoms (delusions, hallucinations, disordered speech, and behavior disturbances) and negative symptoms (avolition, alogia, and anhedonia)[4,5]. However, schizophrenia has considerable overlap with other neurological disorders (e.g., bipolar disorder, autistic spectrum disorder, and Huntington's disease) at both the clinical and genetic levels[6–8], which makes accurate diagnosis quite challenging. Identifying the neural mechanisms of schizophrenia and linking neural signatures to multidimensional clinical symptoms are promising approaches for developing more effective and individual-specific diagnoses.

Noninvasive neuroimaging technology advances the investigation of cognition and brain disorders on the whole-brain scale. The brain has been modeled as a complex network wherein regions of interest (ROIs) are set as nodes, while the functional connections measured by correlation or synchronization between regional signals are edges. Schizophrenia has been widely regarded as a dysconnectivity disorder[9]. In general, schizophrenia is characterized by overall reductions in functional connectivity (FC) compared to that of healthy controls[9–11], as well as alterations in network topologies, including a decline in global efficiency, decreased functional integration, reduced modular structure, and increased global network robustness[12–15]. Abnormal connectivity can predict the total score of schizophrenia and explain part of its neural mechanism, but opposite results have been widely reported[16–18]. For example, the total score of schizophrenia was positively correlated with hyperconnectivity involving the thalamus and temporal cortices[19] but negatively related to hyperconnectivity within the frontoparietal network[16]. While the total score is an overall measure of positive and negative symptom dimensions, the possible reason for these inconsistent observations may be that positive and negative symptoms have separate mechanisms. The dopamine hypothesis, which is currently a widely accepted hypothesis of schizophrenia, notes that enhanced dopamine release may ascribe 'aberrant salience' to irrelevant stimuli (e.g., via failure of top-down inhibitory control) and result in positive symptoms, while reduced release induces a failure to appropriately respond to meaningful reward cues and thus results in negative symptoms[20,21]. Opposing predictions regarding positive and negative symptoms were found for primary motor and cerebellar connectivity[22,23]. Crucially, both the self-similarity and the multifractality of resting-state brain signals were associated with increased negative and positive symptoms, but they had opposite distribution patterns across the brain[24]. Thus, positive and negative symptoms may have opposite effects on the dominant components of the brain FC network.

To address this question, neural signatures that link the brain to diverse schizophrenia symptoms must be explicitly defined. Sufficiently segregated processing in specialized systems and effective global integration are the two basic principles that the brain needs to generate diverse cognitive functions[25]. Abnormalities in segregation and integration have been linked to many brain disorders[25,26], including schizophrenia[27,28]. However, whether positive and negative symptoms have opposite effects on segregation and integration remains unknown. Recently, a nested-spectral partition (NSP) method based on eigenmodes was proposed to detect hierarchical modules in brain networks and describe segregation and integration across multiple levels[29], different from the classical graph measures (e.g., modularity and participant coefficient) at a single level[30]. More importantly, the NSP method has been found to yield better neural signatures than graph theory for linking brain features to cognitive functions and attention-deficit/hyperactivity disorder (ADHD) symptoms[31–33]. It is thus expected that an NSP-based analysis may better reveal the opposite neural biomarkers that underlie positive and negative symptoms in schizophrenia.

Therefore, in this work, we studied hierarchical segregation and integration in brain FC networks in the resting state and explored their distinct associations with positive and negative symptoms in schizophrenia. In addition, the brain dynamically and flexibly integrates neural information across distinctly segregated systems. It has been suggested that dynamic analysis may provide more informative measures related to various neurological and psychiatric conditions, including schizophrenia[19]. We thus constructed dynamic FC networks using open-source functional magnetic resonance imaging (fMRI) datasets derived from schizophrenia patients (SCH group, $n = 50$) and healthy controls (HC group, $n = 50$), and we further defined the strength and variability of dynamic integration/segregation processes. We first studied the alterations in dynamic segregation and integration related to schizophrenia. Second, we identified the associations of dynamic networks with diverse schizophrenia symptoms in positive and negative symptom dimensions. Then, we constructed a machine-learning model to predict positive and negative symptom scores and investigated the opposite effects of positive and negative symptoms on brain FC networks. Finally, we extracted the genes related to distinct effects of positive/negative symptoms and performed Gene Ontology (GO) enrichment analysis.

## Results

**Analysis of dynamic segregation and integration.** Brain functional organization dynamically switches between segregated and integrated states[32]. The NSP method can extract the separated segregation and integration components from a single brain FC network without any threshold, which is able to study the network dynamic transitions in two separate dimensions. When all regions had strongly cooperative activation with high FC (Fig. 1a), the brain network was in an integrated state; otherwise, the connectivity was low, and the brain was in a segregated state. These state transitions can be effectively detected by the dynamic integration and segregation components. We first investigated the integration strength $H_{In}$ and segregation strength $H_{Se}$ during the dynamic state transitions, as measured by time-averaged segregation and integration components. Compared to the healthy control (HC) group, the schizophrenia (SCH) group had decreased integration strength (ANCOVA, $t(98) = -2.845$, $p = 0.004$, Fig. 1b) and increased segregation strength ($t(98) = 4.558$, $p < 0.001$, Fig. 1c). At the local scale, all functional systems had significantly decreased integration strength $H_{In}$ and increased segregation strength $H_{Se}$ (all $p < 0.05$, FDR corrected, Fig. 1d), and the limbic system had the largest alterations (permutation test, $p < 0.001$, see Fig. 1d and Supplementary Fig. 1). We further defined the standard deviations of temporal-resolved integration and segregation components to study the dynamic integration variability $F_{In}$ and segregation variability $F_{Se}$. Integration variability $F_{In}$ in schizophrenia patients was significantly decreased in the whole brain ($t(98) = -3.368$, $p = 0.001$, Fig. 1e), and segregation variability $F_{Se}$ is significantly increased ($t(98) = 5.317$, $p < 0.001$, Fig. 1f), reflecting the more variable segregation process and more stable integration process. These changing trends were observed in all functional systems (all $p < 0.05$, FDR corrected, Fig. 1g), and the limbic system also had a higher decrease in integration variability $F_{In}$ and an increase in

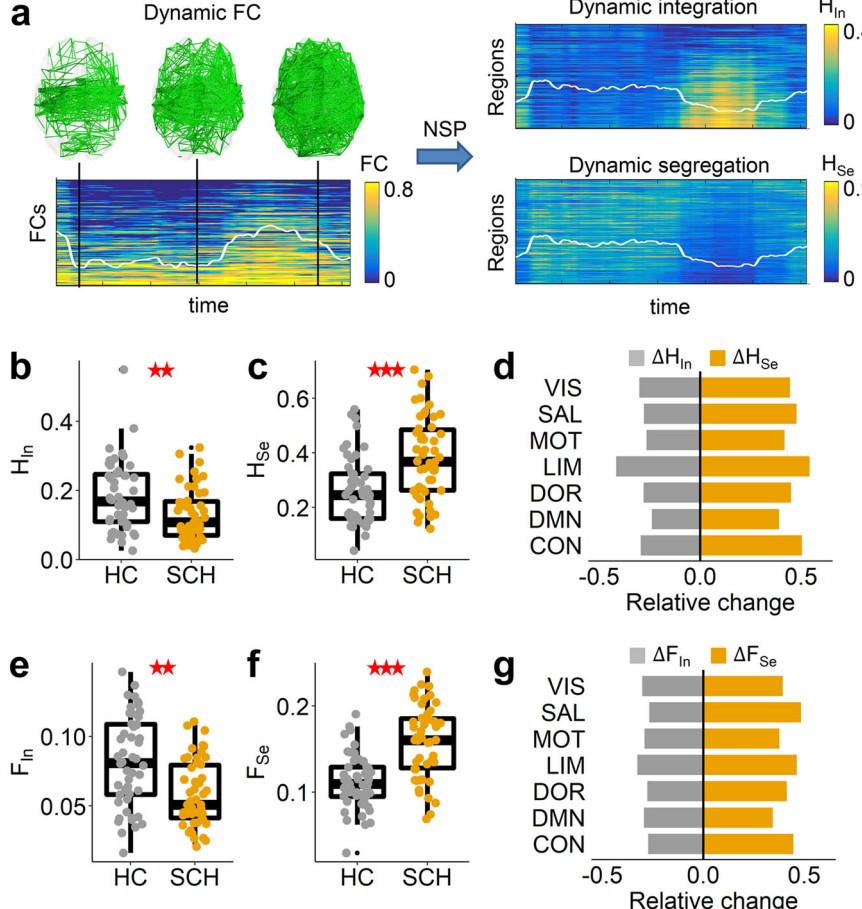

**Fig. 1 Dynamic segregation and integration. a** Schematic diagram of dynamic segregation and integration analysis. The brain functional connectivity (FC) network temporally switches between segregated (low FC) and integrated (high FC) state. Using the NSP method, this dynamic transition can be studied separately in two dimensions: dynamic segregation and integration. The white lines indicate the mean values, and the ranges of the y-axis are indicated by the color bars. **b** Comparisons of integration strength $H_{In}$ and **c** segregation strength $H_{Se}$ between the healthy control (HC, $n = 50$) and schizophrenia (SCH, $n = 50$) groups. Each point indicates the value of an individual, and the boxplot of each group was also provided. ** indicates a significant group difference with $p < 0.01$, *** $p < 0.001$. **d** Relative changes in integration strength $H_{In}$ and segregation strength $H_{Se}$ from HC to SCH in seven functional systems. All systems had significant alterations ($p < 0.05$, FDR corrected). **e** Comparisons of integration variability $F_{In}$, **f** segregation variability $F_{Se}$ between HC and SCH groups. **g** Relative changes in the integration variability $F_{In}$ and segregation variability $F_{Se}$ from HC to SCH in seven functional systems. All systems had significant alterations ($p < 0.05$, FDR corrected). VIS: visual, MOT: motor, DOR: dorsal attention, SAL: salient attention, LIM: limbic, CON: control, DMN: default-mode network (see similar results for the brain parcellation of 500 regions in Supplementary Fig. 3).

segregation variability $F_{Se}$ (see Fig. 1g and Supplementary Fig. 1). We also calculated the corresponding graph theory measures: degree, participant coefficient, variability of degree and variability of participant coefficient, and found that there were no significant differences in graph theory measures between the HC and SCH groups (see Supplementary Fig. 2).

Overall, schizophrenia, characterized by the loss of effective global integration, is related to a more stable global integration process and a more variable segregation process, and the limbic system is most significantly changed. These alterations can be effectively detected by the NSP method relative to graph theory analysis.

**Associations between dynamic brain networks and clinical symptoms in schizophrenia.** We next tested whether the NSP method could link dynamic brain networks to clinical symptoms of schizophrenia. There are a total of ten symptom scores in the positive and negative symptom dimensions, and we focused on the key positive symptom (i.e., hallucinations) and key negative symptom (i.e., avolition). A significant correlation was found for the hallucinations (see Fig. 2a, b and Supplementary Data. 1).

Specifically, the hallucinations are positively correlated with the integration strength $H_{In}$ in the dorsal attention system ($t(48) = 2.205$, $r = 0.303$, $p = 0.032$, Fig. 2a and Supplementary Fig. 4) and negatively related to the segregation strength $H_{Se}$ in the salient attention system ($t(48) = -2.336$, $r = -0.319$, $p = 0.024$) and the dorsal attention system ($t(48) = -2.101$, $r = -0.290$, $p = 0.041$) (see Fig. 2b and Supplementary Fig. 4). A significant correlation was also found for avolition (see Fig. 2c and Supplementary Data. 1). The dynamic segregation variability $F_{Se}$ was negatively related to the avolition score in the DMN ($t(48) = -2.682$, $r = -0.361$, $p = 0.010$) and control systems ($t(48) = -2.367$, $r = -0.323$, $p = 0.022$) (see Fig. 2c and Supplementary Fig. 4). The multivariable regression models further confirmed the above results (see Supplementary Data. 2). Graph theory measures have consistent correlations to the key positive symptoms (i.e., hallucinations) but are not related to negative symptoms (i.e., avolition, see Supplementary Fig. 5)

Overall, brain dynamic networks are related to key positive symptoms (i.e., hallucinations) and key negative symptoms (i.e., avolition). More serious hallucinations correspond to higher integration strength in attention systems. Weaker avolition is

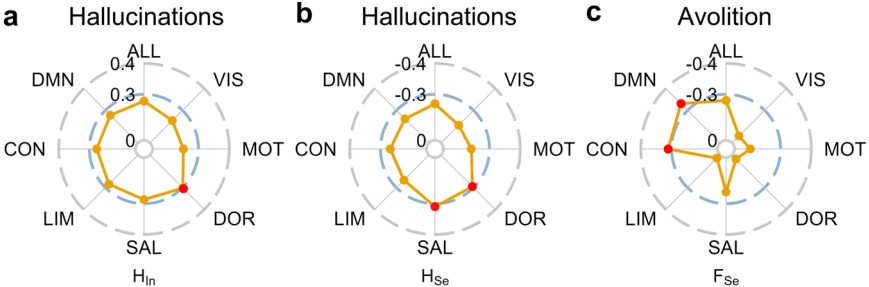

**Fig. 2 Associations between resting-state brain networks and schizophrenia symptoms. a** Correlations of hallucinations score to integration strength $H_{In}$ and **b** segregation strength $H_{Se}$ for the whole-brain (ALL) and seven functional systems. The red dots indicate significant correlations. **c** Correlations between avolition score and segregation variability $F_{Se}$ for the whole-brain (ALL) and functional systems (see Supplementary Data. 1 for more results of correlations between brain network measures and diverse clinical symptoms, see Supplementary Fig. 6 for similar results for the brain parcellation of 500 regions).

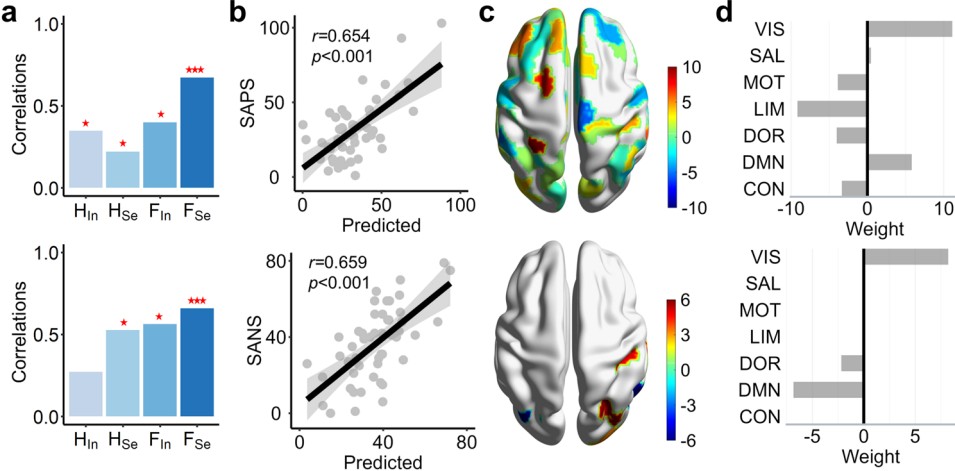

**Fig. 3 Prediction of SANS and SAPS scores. a** Four brain network measures, integration strength $H_{In}$, segregation strength $H_{Se}$, integration variability $F_{In}$, and segregation variability $F_{Se}$, were used to predict the SAPS (upper panel) and SANS (lower panel) scores. The bar charts are correlations between real SAPS/SANS scores and predicted scores using different brain measures. ***$p < 0.001$, **$p < 0.01$, *$p < 0.05$. **b** The correlation between SAPS (upper panel)/SANS (lower panel) scores and predicted scores for segregation variability $F_{Se}$. **c** The weights of regions in the best predictions of SAPS (upper panel)/SANS (lower panel) scores were mapped to the brain surface. **d** The summed weight of the regions in seven functional systems in the best predictions of SAPS (upper panel)/SANS (lower panel) scores (see similar results for the brain parcellation of 500 regions in Supplementary Fig. 8).

associated with higher dynamic variability of the segregation process in the default-mode network (DMN) and control systems.

**Predictions of positive and negative symptoms.** In the clinic, the negative and positive symptom scores (SANS and SAPS) are crucially important to diagnose schizophrenia, but we did not find a significant correlation between these two scores and brain network measures (see Supplementary Table. 1). To further explore the neural mechanisms of schizophrenia, we utilized a machine-learning approach to predict SANS and SAPS scores. Linear regression models were constructed, and leave-one-out cross-validation (LOO-CV) was used to validate the results. In the prediction model, the input features were regional measures, including integration strength $H_{In}^i$, segregation strength $H_{Se}^i$, integration variability $F_{In}^i$, and segregation variability $F_{Se}^i$. The prediction accuracy was estimated by the correlation between real and predicted scores that was further tested by the permutation test (10000 times). Our results showed that segregation variability $F_{Se}$ has the best prediction of the SAPS score ($r = 0.654, p<0.001$) and SANS score ($r = 0.659, p<0.001$) (Fig. 3a, b) and outperforms the classical graph theory measures (see Supplementary Fig. 7). By summing the weights of regions belonging to the same system (Fig. 3c), we studied which system predominantly

contributed to the SAPS/SANS scores. In the SAPS prediction model, the visual and DMN systems had high positive weights, and the limbic system had a high negative weight (Fig. 4d). Thus, the more variable segregation in the limbic system contributes to smaller SAPS, but higher variability in the visual and DMN systems corresponds to higher SAPS. In the SANS prediction model, the featured regions were distributed in the visual and DMN systems. The visual system had a high positive weight, and the DMN system had a high negative weight (Fig. 4d), indicating that more variable segregation in the DMN system contributes to a smaller SANS, but the opposite is true for the visual system.

Overall, compared to graph theory measures, NSP-based features can better predict the SAPS and SANS scores, suggesting potential biomarkers for schizophrenia. The DMN, visual and limbic systems significantly participate in the predictions, reflecting their important roles in schizophrenia.

**Opposite effects of positive and negative symptoms on the brain.** In Fig. 3d, the DMN system has opposite contributions to the predictions of SANS and SAPS scores, reflecting that SANS and SAPS may have contrary effects on brain functional organizations. However, it should be noted that SANS and SAPS scores are positively correlated ($r = 0.261$, $p = 0.067$,

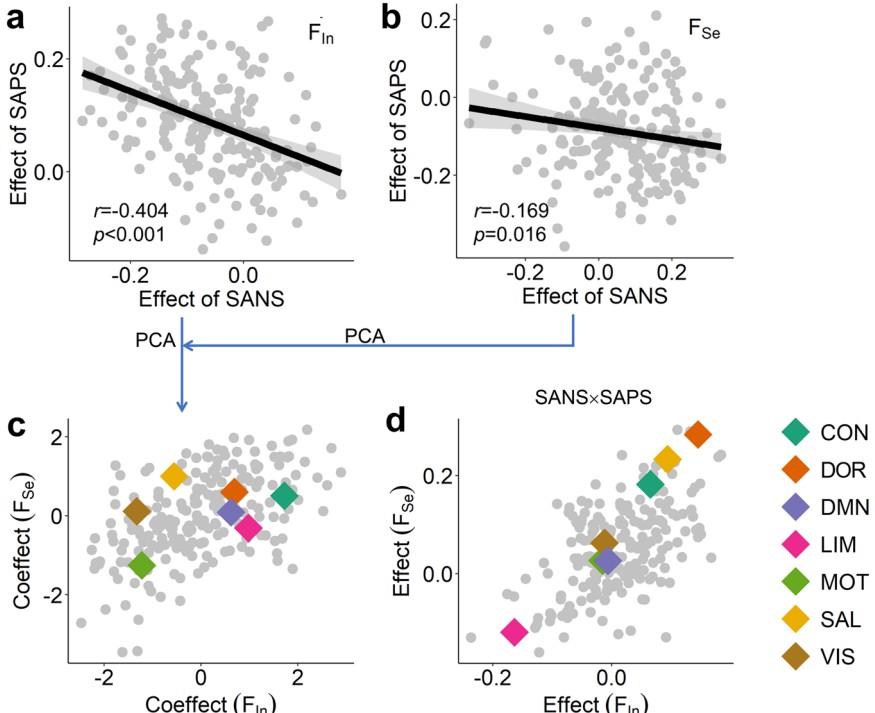

**Fig. 4 Effects of schizophrenia symptoms on brain networks.** Effects of SANS/SAPS on **a** dynamic integration variability $F_{In}$ and **b** segregation variability $F_{Se}$. The effects were measured by the regression coefficients in the multiple regression model (Eq. 8). **c** Coeffects of SANS/SAPS on integration variability $F_{In}$ and segregation variability $F_{Se}$. The coeffects were obtained from **a** and **b** using principal component analysis (PCA). The averaged coeffects of SANS and SAPS on regions within seven functional systems are marked. **d** Interaction effect of SANS and SAPS on integration variability $F_{In}$ and segregation variability $F_{Se}$. The coeffects on the seven functional systems are also shown (see similar results for the brain parcellation of 500 regions in Supplementary Fig. 10).

$t(48) = 1.872$), indicating the existence of a latent common domain between them. To test the above possibilities, we constructed a multiple regression model to evaluate the effects of SANS and SAPS on brain networks, as well as their interaction effect (Eq. 8). We indeed found that the effect of SAPS and the effect of SANS on regions are negatively correlated for dynamic segregation variability $F_{Se}$ ($t(48) = -2.419$, $r = -0.169$, $p = 0.016$) and integration variability $F_{In}$ ($t(48) = -6.211$, $r = -0.404$, $p < 0.001$, see Fig. 4a, b). More importantly, most of the SANS effect is negative for integration variability $F_{In}$ among regions, and most of the SAPS effect is positive; for segregation variability $F_{Se}$, most of the SANS effect is positive, and most of the SAPS effect is negative, indicating that the SANS and SAPS have opposite effects on the dynamic variability of integration and segregation in brain networks. These opposite effects of SAPS and SANS on regions were also found in graph theory measures (see Supplementary Fig. 9).

To identify which systems are predominantly affected by SANS or SAPS, we performed principal component analysis (PCA) on the two vectors of effects of SANS/SAPS for integration variability $F_{In}$ and segregation variability $F_{Se}$, respectively. The first component was used to measure the coeffect by SANS and SAPS (explained 70.19% for integration variability $F_{In}$ and 58.47% for segregation variability $F_{Se}$), wherein positive values indicate the effect of SANS and negative values indicate the effect of SAPS (Fig. 4c). We then averaged the coeffects of regions within each system to represent the coeffects on this system. The motor system has a consistently high negative coeffect for both integration variability $F_{In}$ and segregation variability $F_{Se}$, indicating that SAPS predominantly affects the dynamic segregating and integrating process in this system. However, the largest positive coeffect is in the control system for integration variability $F_{In}$ and

in the salient attention system for segregation variability $F_{Se}$. Thus, SANS dominantly affects the integration variability $F_{In}$ of the control system and the segregation variability $F_{Se}$ of the salient attention system. We further studied the interaction effects of SANS and SAPS on brain networks. The regression coefficients of the SANS × SAPS item of regions were also averaged within each system to represent the interaction effect on this system. Dorsal and salient attention systems have the highest interaction effect for both integration variability $F_{In}$ and segregation variability $F_{Se}$ (Fig. 4d), indicating that SANS and SAPS interactively affect both segregating and integrating processes in attention systems.

Overall, positive and negative symptoms of schizophrenia have opposite effects on the functional organization of resting-state brains. SAPS predominantly affects the motor system; SANS dominantly affects the control system; SANS × SAPS interaction interactively affects both attention systems.

**Linking effects of schizophrenia symptoms to gene transcriptional profiles.** To investigate whether the components dominantly affected by SAPS/SANS are related to clinical manifestations, we first linked the effects of regions to gene transcriptional profiles and then performed gene ontology (GO) enrichment analysis. We collected whole-brain gene expression data from the Allen Human Brain Atlas and mapped the gene expression profiles to 200 parcellation regions (see Methods). Each region contains 15,633 genes (Fig. 5a). To identify the significant genes, the coeffects of SANS/SAPS on integration variability $F_{In}$ and segregation variability $F_{Se}$ (see Fig. 4c) were gathered to perform a PCA. The first component (explaining 76.46% of the variance) was extracted to reflect the overall effect of schizophrenia (Fig. 5a), wherein high positive values represent

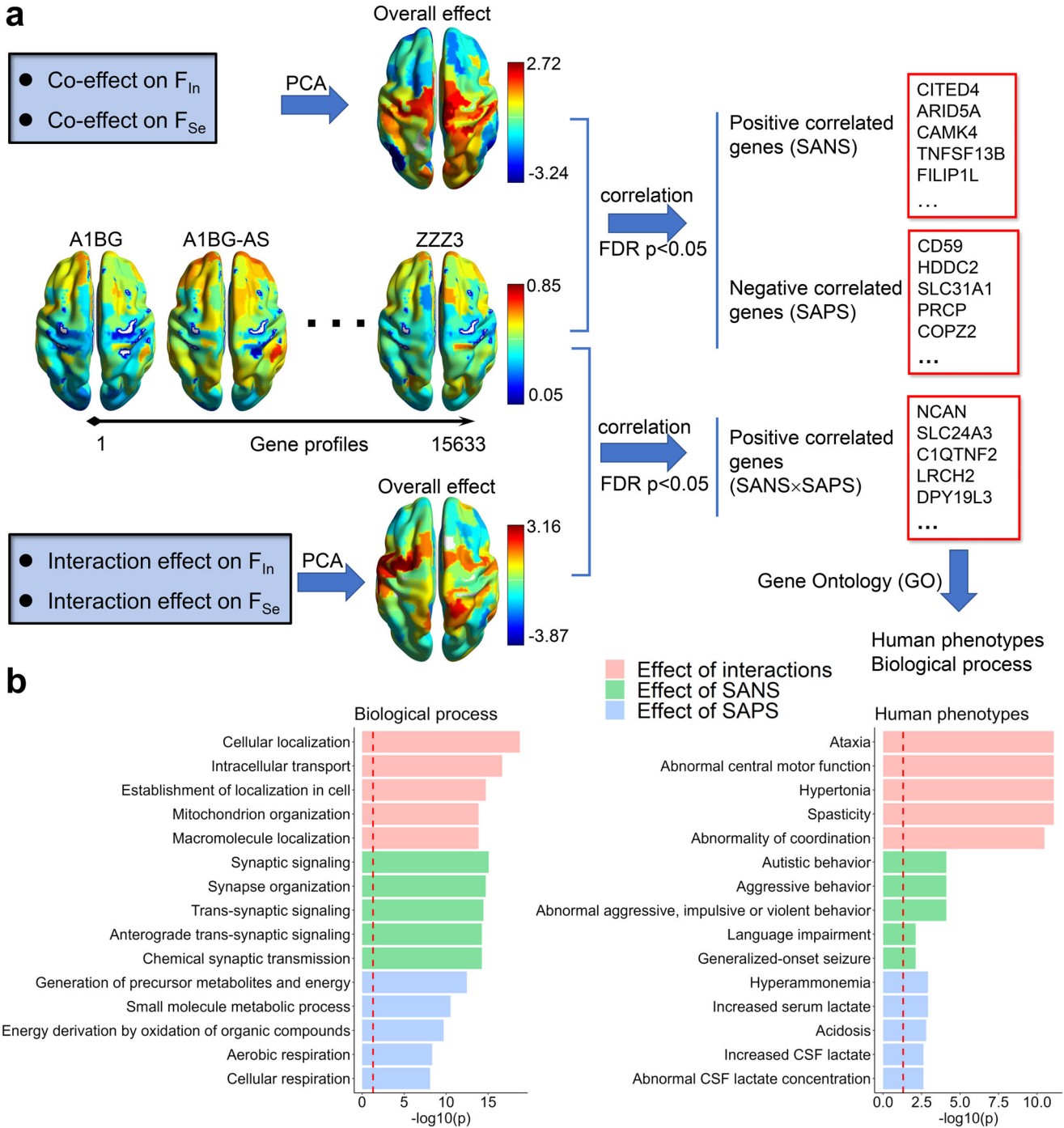

**Fig. 5 Effects of schizophrenia symptoms and gene profiles. a** The pipeline of gene selection for SANS, SAPS, and SANS × SAPS interaction effects. The correlations between the overall effect and gene transcriptional profiles were calculated to identify significant genes ($p < 0.05$, FDR corrected). **b** Top five gene ontology (GO) enrichment results for biological processes and human phenotypes using ToppGene, showing specific functional relevance (see similar results for the brain parcellation of 500 regions in Supplementary Fig. 11).

the overall effect of SANS, and high negative values represent the overall effect of SAPS. By calculating the correlations between the overall effect and gene transcriptional profiles, we identified 1780 SANS-related genes that had significant positive correlations ($p < 0.05$, FDR corrected) and 2281 SAPS-related genes that had significant negative correlations ($p < 0.05$, FDR corrected). We then used these significant genes to perform GO annotation analysis using the ToppGene Suite. The overall effect of SANS is mainly related to biological signaling through synapses, and the corresponding human phenotypes include abnormal autistic,

aggressive, and violent behavior (Fig. 5b). The overall effect of SAPS is related to the generation of precursor metabolites and energy and to small molecule metabolic processes. The corresponding human phenotypes include hyperammonemia, increased serum lactate, and acidosis (Fig. 5b).

Furthermore, we also focused on whether the components affected by SAPS × SANS are related to clinical manifestations. In the PCA on the effects of SANS×SAPS on integration variability $F_{In}$ and segregation variability $F_{Se}$ (see Fig. 4d), the first component (explained 77.19% of variance) was extracted to

reflect the overall effect of interactions (Fig. 5a), wherein high positive values indicate the high overall effect by the interaction. We identified 3055 SANS × SAPS-related genes that had significant positive correlations with the overall effect of the interaction ($p < 0.05$, FDR corrected). GO analysis showed that the SANS × SAPS interaction is mainly related to biological cellular localization and intracellular transport, and the corresponding human phenotypes are ataxia, abnormal central motor function, and hypertonia (Fig. 5b). Moreover, schizophrenia-related genes (27/32 available in the Allen data) with damaging ultrarare mutations have been identified[34]. We observed 12 genes associated with the effects of SANS and SAPS (see Supplementary Table. 2), and nine genes were related to the SANS × SAPS interaction, indicating the correspondence between ultrarare mutation genes and common effects of schizophrenia symptoms. For graph theory measures, the number of significantly related genes is not enough to statistically enrich biological processes and human phenotypes; e.g., only 194 SANS-related genes were identified for the graph theory measures.

Overall, the effect of SANS is related to synaptic signaling processes and abnormal autistic, aggressive, and violent behaviors; the effect of SAPS is related to metabolic precursors, hyperammonemia, and acidosis; and the SANS × SAPS interaction effect is related to cellular localization and transport and abnormal motor function. All these phenotypes have been found in schizophrenia[35–39].

## Discussion

To identify the neural mechanisms of diverse clinical symptoms in schizophrenia, we studied dynamic functional segregation and integration based on hierarchical modules in brain FC networks. The brain networks of schizophrenia patients show a loss of effective global integration, accompanied by a more stable dynamic integrating process and a more variable segregating process. The limbic system has the highest relative degree of alteration. Additionally, among positive symptoms, hallucinations are linearly related to increased network integration in the dorsal attention system; among negative symptoms, avolition is negatively related to segregation variability in the DMN and control systems. Furthermore, by adopting a machine-learning approach, we successfully predicted the SANS and SAPS scores and suggested that NSP-based dynamic features may be biomarkers for schizophrenia. Most importantly, SANS and SAPS scores have opposite effects on integration/segregation variabilities in brain networks. The GO enrichment analysis revealed that the effect of SANS was related to autistic, aggressive, and violent behaviors; the SAPS effect was associated with hyperammonemia and acidosis; and the SANS×SAPS interaction was correlated with abnormal motor function. These results have potential applications for further separately identifying the positive and negative symptoms and could thus contribute to the development of more accurate diagnostic criteria for schizophrenia.

Schizophrenia has been regarded as a dysconnectivity disorder[9]. While many studies have found decreased integration at the whole-brain level in schizophrenia[9–15], we further confirmed that schizophrenia is typically characterized by the loss of global integration. In contrast to a previous study in which schizophrenia patients showed consistently increased resting-state FC variability in the sensory and perceptual systems and decreased variability in high-order networks at the region and network levels[40], we used the NSP method, which separates FC into segregation and integration components, and provided that the dynamic integration process is more stable in schizophrenia patients than in healthy controls and that the dynamic segregation process is more variable. Thus, the brains of schizophrenia

patients may lose the flexibility to integrate and coordinate different neural systems in response to internal and external stimuli. Interestingly, the limbic system has the largest alterations in all network measures for schizophrenia patients, which has long been implicated in the pathogenesis of schizophrenia[41,42] and displays significantly greater deformation[43]. This system is believed to drive disruptions in both the extrinsic (i.e., delusions) and intrinsic (i.e., hallucinations) interpretation of sensory stimuli in positive symptoms[44], and it has also been suggested to drive affective flattening and change in the sense of self in negative symptoms[45,46]. In fact, the limbic system involves a set of regions in the paleocortex and supports a variety of functions related to emotion regulation and motivation meditation[47,48]. The development of the limbic system reduces impulsive choices from early adolescence to mid-adulthood[49], and abnormal immaturity of this system confidently predicts hyperactivity[50]. In particular, the limbic system best predicts the increase in hyperactivity in ADHD patients across the lifespan[33]. Here, the SANSxSAPS interaction has the largest negative effect on the limbic system, and its GO-based human phenotype is related to motor dysfunction that has been widely observed in schizophrenia[35–38,51,52]. Our results provide further support that the limbic system is a key factor affecting individuals with schizophrenia[42] and may underlie abnormal motor function.

Hallucinations, a core positive symptom of schizophrenia, have been regarded as a failure of the top-down suppression of bottom-up perceptual processes[21]. Abnormal attribution of salience to external and internal stimuli is a core feature of schizophrenia[53]. We found that higher integration in salient and dorsal attention systems is closely related to several hallucinations. The salient attention system was thought to enable brains to direct attention toward salient stimuli by excluding irrelevant noise, which supports automatic "bottom-up" forms of attention[54], and the dorsal attention network is engaged to exert top-down influences on visual areas during the spatial orienting of attentional tasks, which was greater than the reverse bottom-up effects from the visual cortex[55,56]. Both systems are typically task-positive, and their abnormalities are related to the brain imbalance between top-down and bottom-up controls[57] that may underlie the impaired hallucinations[58] in terms of functional organization and structural anatomy[1,53,58,59]. Our results further reinforce the importance of the salient attention system in schizophrenia[27]. Meanwhile, many studies have proposed that hallucinations may arise with dysconnectivity of the salient attention system with other systems[27], especially with the DMN[27,58,60]. Here, we did not find a close relationship between the DMN and hallucinations, as observed in previous studies[58,60], e.g., strong FC within the DMN and spontaneous DMN withdrawal for the hallucination state[58,60], but we found that a less variable segregation process in the DMN and control systems is related to more severe avolition, which is the core negative symptom in schizophrenia[61] and reflects a reduction in the motivation to initiate or persist in goal-directed behavior. The DMN and control networks are associated with goal-directed behavior[62], and their abnormalities are closely related to avolition[63–65], as confirmed in our study. Meanwhile, the salience-monitoring theory proposes that abnormal coupling between the salient attention system and DMN begets positive and negative symptoms of schizophrenia[66]. Our relationships between the DMN and avolition and between the salient attention system and hallucinations provide further support for this hypothesis.

We did not find a significant relationship between SANS/SAPS scores and brain features, including both NSP and graph theory measures, at the whole-brain level or the system level, but we adopted the machine-learning method to successfully predict the

scores. Compared to classical graph theory, the NSP-based method had superior performance in predicting SANS/SAPS scores and detecting network alterations, reflecting the advantages of our method based on hierarchical modules in brain FC networks. This result is highly consistent with a series of our works wherein the NSP-based method is more powerful in linking the brain to diverse cognitive abilities[32], task performance[31], stress conditions[67], ADHD symptoms[33], and bipolar disorder symptoms[68]. All these findings demonstrated that the NSP-based features detected across multiple levels are promising biomarkers for schizophrenia and other brain disorders.

In the prediction models, the features in the DMN have opposite contributions to SAPS and SANS, and using a multivariate regression model, we further confirmed that SANS and SAPS have opposite effects on brain networks. Previous studies found that primary motor and cerebellar connectivity have opposite predictions on positive and negative symptoms[22,23], and self-similarity and multifractality of resting-state brain signals with opposite distribution patterns have the same associations with negative and positive symptoms[24]. Here, we provided the first direct evidence that positive and negative symptoms of schizophrenia have opposite effects on the functional organization of resting-state brains while excluding their interaction, especially that the DMN has opposite contributions to the predictions of SANS and SAPS scores. The DMN is negatively correlated with other systems[69], and its abnormality may underlie the positive symptoms[58,60,70] and negative symptoms[63–65] of schizophrenia. Hare SM et al. reported that FC with a 4 s lag between the anterior DMN and posterior DMN was negatively associated with the severity of disordered thought and attentional deficits, and FC with a 2 s lag between the anterior DMN and salience network was positively related to the severity of flat affect and bizarre behavior[66], which is highly consistent with our observations of opposite functions of the DMN on positive and negative symptoms. In particular, negative symptoms often persist after treatment with antipsychotic medication[71]. Even though negative symptoms (e.g., anhedonia/asociality) were found to be related to the posterior cingulate and precuneus, part of the DMN, in a two-tone auditory oddball task, identifying reliable targets of regions for treatment remains a challenge in the clinic[72]. Our results greatly extend the understanding of schizophrenia and provide that distinct DMN regions may be targets for positive and negative symptoms.

Using GO enrichment, we demonstrated that the SANS × SAPS interaction is related to the pathology of intracellular transport and cellular localization, such as in mitochondria[35,52]. These biological processes may impact neuronal development, synaptic function, and plasticity[36,37]. The corresponding phenotypes are related to ataxia and abnormal motor function, which have been widely observed in schizophrenia[38,51]. In particular, visual control influenced the age-associated increase in ataxic gait[51], and we found that the visual system contributed to both positive and negative symptoms in the machine-learning prediction models. These results may suggest baseline pathological changes in motor function in schizophrenia. Meanwhile, we confirmed that 12/32 schizophrenia-related genes had damaging ultrarare mutations[34], and 9/12 genes were related to the SANS × SAPS interaction, indicating that ultrarare mutation genes may mainly contribute to the baseline symptoms of abnormal motor function in schizophrenia. Thus, beginning with motor abnormalities, further development of the disorder in different directions may generate positive and negative symptoms. More specifically, negative symptoms may be inherent to the alternated biological process in synapses that transfer neural information between neurons, as also suggested by the genetics and protein-interaction evidence for the role of

postsynaptic signaling processes in schizophrenia[36,73]. In human phenotypes, negative symptoms are related to autistic and aggressive behaviors that extensively overlap between schizophrenia[38] and autism[74]. As language disturbances are a key feature of schizophrenia[75], our results suggest that patients are unable to flexibly communicate with others and effectively express themselves, resulting in impulsive and violent behaviors in the clinic, as also seen in autism[76]. Finally, we found that positive symptoms are related to the abnormal biological process of metabolism and the phenotypes of hyperammonemia, increased serum lactate, and acidosis. A recent meta-analysis on lactate or pH in schizophrenia revealed a significant increase in lactate in schizophrenia and a nonsignificant decrease in pH[77]. Our GO enrichment results provide further evidence that abnormal metabolic processes in schizophrenia brains result in the accumulation of ammonia, inducing hyperammonemia, acidosis, and increased serum lactate[39]. All these abnormalities are closely related to schizophrenia[39,78], especially acidosis altering dopamine and glutamate neurotransmission, causing symptoms of schizophrenia[39].

## Methods

**Participants**. The dataset was extracted from the UCLA Consortium for Neuropsychiatric Phenomics LA5c Study[79]. Fifty schizophrenia patients (female: 12; age: 36.46 ± 8.88 years old) and 50 healthy controls (female; 12, age: 34.84 ± 9.03 years old) were included. There was no significant difference in age (two-sample $t$-test, $t(98) = 0.905$, $p = 0.354$). The clinical symptoms of schizophrenia patients were evaluated with the Scale for the Assessment of Positive Symptoms (SAPS) and the Scale for the Assessment of Negative Symptoms (SANS)[80]. The SANS includes five symptom dimensions, namely, avolition, alogia, anhedonia, attention, and affective flattening; the SAPS includes hallucinations, delusions, bizarre behavior, thought disorder, and blunted affect. The clinical scores of these symptoms are provided in the Supplementary Data. 3 (SANS) and Supplementary Data. 4 (SAPS). All studies were conducted in accordance with principles for human experimentation as defined in the Declaration of Helsinki and the International Conference on Harmonization Good Clinical Practice guidelines. All participants gave written informed consent according to the procedures approved by the University of California Los Angeles Institutional Review Board.

**MRI data processing**. Each participant completed one resting-state fMRI scanning session (time of repetition [TR] = 2 s), lasting for 304 s (152 frames); see ref. [81] for more detailed scanning parameters. Resting-state fMRI data were processed using FSL (http://www.fmrib.ox.ac.uk/fsl/) and AFNI (http://afni.nimh.nih.gov/afni/) software in the Ubuntu 14.04 system[29]. The procedure included (1) slice-timing correction to the median slice; (2) motion correction; (3) segmenting the anatomical image; (4) Montreal Neurological Institute (MNI) normalization; (5) spatial smoothing using a Gaussian kernel with a 6-mm full width at half maximum (FWHM); (6) bandpass filtering (0.01–0.1 Hz); and (7) elimination of 6 rigid body motion correction parameters and the signal from the white matter and a ventricular region of interest using linear regression. The mean framewise displacement (FD) was 0.160 ± 0.159 mm for the healthy control group and 0.267 ± 0.215 mm for the schizophrenia group. The difference in FD between the two groups was significant (two-sample $t$-test, $t(98) = 2.779$, $p = 0.004$). Thus, an analysis of covariance (ANCOVA) was carried out for the group comparison. Since the global whole-brain signal was related to brain network integration and segregation (Supplementary Table. 3) and may contain the clinical information of schizophrenia symptoms[82–84], it was not removed from our analysis.

**Brain functional connectivity**. The brain was parcellated into $N = 200$ regions of interest (ROIs) using the Schaefer atlas[85], and the results were similar for the brain parcellation of 500 regions (see Supplementary Figs. 3, 6, 8, 10, 11). The blood oxygen level-dependent (BOLD) signals of voxels within each region were averaged to obtain the regional fMRI time series, and the Pearson correlation coefficient was used to estimate the FC between regions. The BOLD signals were divided into pieces using the sliding window method, and temporal-dynamic FC was calculated in each window. As suggested by ref. [86], we chose a window width of 60 s (30 points) and a sliding step of 2 s (1 point), and there were 132 windows. Meanwhile, group-stable, individual static FC networks were also constructed, which were used to address the limitation of shorter fMRI series lengths resulting in stronger network segregation[32] (see fMRI length calibration). For the group-stable FC, the fMRI time series for all participants in each group were concentrated, and the FC was computed on a sufficiently long time scale. Individual static FC networks were constructed using the whole fMRI time series in each participant. In all FC

networks, negative connectivity was set to zero, and the diagonal elements were kept at one[32,87,88].

**Nested-spectral partition (NSP) method**. The NSP method was introduced to detect hierarchical modules in FC networks based on eigenmodes. The FC matrix C can be decomposed into functional modes with eigenvectors $U$ and eigenvalues $\Lambda$, and the modes were sorted according to the descending order of eigenvalues $\Lambda$. The NSP method has the following procedures[32]:

1. In the first functional mode, all regions had the same negative or positive eigenvector value; this mode was regarded as the first level, with one module (i.e., whole-brain network).
2. In the second functional mode, the regions with positive eigenvector signs were assigned to a module, and the regions with negative signs formed the second module. This mode was regarded as the second level, with two modules.
3. Based on the positive or negative sign of regions in the third mode, each module in the second level was further partitioned into two submodules, forming the third level. Subsequently, the FC network could be modularly partitioned into multiple levels with the order of functional modes increasing (see Supplementary Fig. 12 for a more detailed description of the process). Regions within a module in a level may have the same sign of eigenvector values in the next level, and then the module is indivisible, which has no effect on the subsequent partitioning process. When each module contained only a single region at a given level, the partitioning process was stopped.

After the partitioning process, the NSP method outputs the module number $M_i(i = 1, \cdots, N)$ and the modular size $m_j(j = 1, \cdots, M_i)$, e.g., the number of regions within a module, at each level.

**Hierarchical segregation and integration components**. Functional segregation and integration in brain FC networks were defined across hierarchical modules that were detected by the NSP method[31,32]. Consistent with the graph-based modularity[30], modules at a given level support the segregation between them and integration within them. The increased module number $M_i$ with the increasing order of functional mode reflects higher segregation. At each level, segregation and integration can be defined as[32]:

$$H_i = \frac{\Lambda_i^2 M_i(1 - p_i)}{N} \tag{1}$$

with

$$p_i = \frac{\sum_j |m_j - N/M_i|}{N} \tag{2}$$

Here, $N$ is the number of regions; $\Lambda_i$ is the eigenvalue for the $i$-th functional mode; $p_i$ is a correction factor for heterogeneous modular size and reflects the deviation from the optimized modular size $m_j = N/M_i$ at the $i$-th level. Since the first level contains only a single module for all regions, this level was taken to reflect the global integration component:

$$H_{In} = \frac{H_1}{N} = \frac{\Lambda_1^2 M_1(1 - p_1)}{N^2} \tag{3}$$

With the increasing order of functional modes, the levels contain more modules with smaller sizes and support higher segregation. Thus, the segregation component was summed from the second to $N$th levels:

$$H_{Se} = \sum_{i=2}^{N} \frac{H_i}{N} = \sum_{i=2}^{N} \frac{\Lambda_i^2 M_i(1 - p_i)}{N^2} \tag{4}$$

Consequently, for a single FC network, we obtained the separated integration component $H_{In}$ and segregation component $H_{Se}$. A larger $H_{Se}$ and smaller $H_{In}$ reflect stronger network segregation and weaker global integration.

The contribution of each region to the integration and segregation components can be further defined as:

$$H_{In}^j = H_1 U_{1j}^2 \text{ and } H_{Se}^j = \sum_{2}^{N} H_i U_{ij}^2 \tag{5}$$

where $\sum_{j=1}^{N} U_{ij}^2 = 1$ for the $i$-th functional mode. The integration and segregation of the functional system were obtained by averaging the corresponding components of regions within this system.

For the dynamic FC networks, the time-resolved segregation component $H_{Se}(t)$ and integration component $H_{In}(t)$ at each time window for each individual were obtained. The values of integration and segregation strength were defined as the average values of $H_{In}(t)$ and $H_{Se}(t)$ over time, respectively. The values of integration and segregation variability were calculated as follows:

$$F_{In}^j = \sigma_{H_{In}^j} \text{ and } F_{Se}^j = \sigma_{H_{Se}^j} \tag{6}$$

where $\sigma_{H_{In}^j}$ and $\sigma_{H_{Se}^j}$ represent the standard deviations of the $H_{In}^j(t)$ and $H_{Se}^j(t)$ time series.

Notably, finer parcellation of the brain (i.e., 500 regions) would generate more modules with smaller sizes in higher-order levels of brain functional networks, accompanied by a larger segregation component and lower integration component (see Supplementary Fig. 13). However, the results for schizophrenia are robust for different brain parcellations (see Supplementary Figs. 3, 6, 8, 10, 11).

**fMRI length calibration**. Since shorter fMRI series lengths result in stronger apparent network segregation[32], we adopted a proportional calibration scheme to address this limitation[32]. Assume that the integration component of the stable FC network in each group is $H_{In}^S$ and that the integration components of individual static FC networks for all participants are $H_{In} = [H_{In}(1), H_{In}(2), \cdots, H_{In}(50)]$. The group-averaged integration component is calibrated to the stable component:

$$H_{In}'(n) = H_{In}(n) \times H_{In}^S / \langle H_{In} \rangle \tag{7}$$

Here, $\langle \rangle$ represents the group average across all participants, and $n$ represents the individual. Then, calibration was also performed for the regional integration component $H_{In}^j$. For region $j$ of the $n$-th participant, the calibrated regional integration component is $H_{In}^{j'} = H_{In}^j / H_{In}(n) \times H_{In}'(n)$, where the relative contribution of each region to network integration remains consistent.

For dynamic FC networks, the temporal integration component $H_{In}(t)$ for each individual was calibrated to its static integration component $H_{In}'$ to maintain the individual rankings. The vector of the dynamic integration component for an individual across all windows was $h_{In} = [h_{In}^1, h_{In}^2, \cdots, h_{In}^{132}]$, and the calibrated result was calculated as $h_{In}^{t'} = h_{In}^t H_{In}^{i'} / \langle h_{In} \rangle$. Here, $\langle \rangle$ represents the average across time windows.

The same calibration processes were performed for the segregation component on the global and local scales and in static and dynamic networks, and the calibration was performed separately in each group.

**Machine-learning prediction model**. The scikit-learn toolbox was used to construct a machine-learning prediction model[89]. First, we used the function *linear_model. Linear regression* to build linear predictive models. The independent variables were regional measures (i.e., $H_{In}^j$, $H_{Se}^j$, $F_{In}^j$, $F_{Se}^j$), and the dependent variables were the SANS or SAPS scores. Second, leave-one-out cross-validation (LOO-CV) was applied with the function *cross_val_predict*. In each iteration of LOO-CV, one participant was selected as the test set, and the remaining participants were selected as the training set. This process was repeated until every participant had been selected as a test set once. Then, we used the correlation between the real clinical score and the predicted score to evaluate the prediction accuracy, and the statistical comparison was performed by permuting the ranks of clinical scores (10,000 times). In the prediction model, the functions *f_regression* and *SelectKBest* were used to select features. The *f_regression* function calculated the correlations between regional measures and clinical scores and sorted the regions according to their F values. Then, the first $K$ features were selected and fed into the prediction model. Here, we varied $K$ from 1 to $N$ and chose the best $K$, defined as the value at which the model had the best predictive performance. The input features were normalized such that the weights of regions were comparable.

**Effects of SANS and SAPS on the brain**. To extract the effects of positive and negative symptoms, as well as their interaction effect on brain FC networks, we built a multiple regression model:

$$H \sim \text{SANS} + \text{SAPS} + \text{SANS} \times \text{SAPS} + \text{sex} + \text{age} + \text{FD} \tag{8}$$

Here, $H$ is the brain measure for each region, i.e., $H_{In}^j$, $H_{Se}^j$, $F_{In}^j$ and $F_{Se}^j$. The regression coefficients of SANS and SAPS reflect the effects of negative and positive symptoms on the brain, and the coefficient of SANS × SAPS indicates the interaction effect. FD is the mean framewise displacement.

**Gene Ontology (GO) enrichment analysis**. The gene expression data used in this study were extracted from the Allen Human Brain Atlas (AHBA)[90]. This open-source project contains ~3700 tissue samples from six donors and provides the Montreal Neurological Institute (MNI) coordinates of the tissues. The tissue samples from four donors are limited to the left hemisphere, and the samples from the remaining two donors span the whole brain. The *abagen* toolbox was used to map the microarray gene expression data to 200 regions in the Schaefer atlas[91]. This toolbox provides a standardized processing procedure of accepting an atlas and returning a parcellated regional gene expression matrix. Here, we used the default settings, as suggested by ref. [91]. Although only two donors had gene expression data available from the right hemisphere, we chose to use whole-brain gene expression due to the asymmetry between the left and right hemispheres. We calculated the Pearson correlations between gene expression and network components affected by SANS/SAPS scores to identify the significant genes ($p < 0.05$, FDR corrected), which were further processed with ToppGene Suite to perform GO annotation analysis (FDR correction method, significance cutoff level of 0.01).

**Statistics and reproducibility.** Statistical analysis was performed with MATLAB R2016b and R (v4.0.4). A two-sample *t*-test was used to compare the age and FD between the two groups. ANCOVA (analysis of covariance) tested the between-group differences in brain network measures with FD as the confounding variable. FDR method of Benjamini–Hochberg was used for multiple comparisons. A permutation test (1000 times) was conducted to test the differences in relative changes between different systems. Pearson correlation was used to evaluate the relationships between brain network measures and symptom scores. *P* value < 0.05 was considered statistically significant.

To test the reproducibility of results, we used the Schaefer atlas to parcellate the brain into $N = 200$ regions (main analysis) and $N = 500$ regions (reproducibility analysis), and reported consistent results for these two parcellations. We also performed the graph theory analysis for $N = 200$ regions, and the results are also similar.

**Reporting summary.** Further information on research design is available in the Nature Portfolio Reporting Summary linked to this article.

## Data availability

The original MRI and clinical scores datasets are available at https://openneuro.org/datasets/ds000030. The original gene data were available at http://human.brain-map.org/. The brain atlas and the partition of the seven functional systems are available at https://github.com/ThomasYeoLab/CBIG/tree/master/stable_projects/brain_parcellation. The abagen toolbox is available at https://abagen.readthedocs.io/en/stable/index.html#. ToppGene Suite is available at https://toppgene.cchmc.org/.

## Code availability

The code used in this study and the preprocessed gene data were available at https://github.com/TobousRong/schizophrenia.

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

## Acknowledgements

This work was supported by the National Natural Science Foundation of China (No. 12272292 and No. 11802229) and the Natural Science Basic Research Program of Shaanxi (No. 2022JQ-005).

## Author contributions

X.W.: Data processing, formal analysis, writing—original draft, and software. Z.C.: Visualization, writing—review and editing. R.W.: Methodology, funding acquisition, writing—introduction and discussion, writing—review and editing.

## Competing interests

The authors declare no competing interests.
