## [Peer Review File · Communications Biology]

Reviewers' comments:

Reviewer #1 (Remarks to the Author):

Wang et al. studied the brain's dynamic functional segregation and integration in schizophrenia patients, comparing to healthy controls. The authors utilized a nested-spectral partition (NSP) approach to detect hierarchical modules in an open resource fMRI dataset shared by UCLA. It was found that the brain networks of schizophrenic patients had a reduced global integration, accompanied by a more stable dynamic integrating process and a more variable segregating process. Brain network and gene ontology analyses were performed with a package of results. Linking brain networks with distinct positive and negative symptoms in schizophrenia appears to be novel. Overall, the manuscript is well-written. The analytical approach and findings are solid.

I have a few methodological concerns to share.

1. The measures about hierarchical segregation and integration are mostly derived from the authors' previous study (Wang et al., 2021, PNAS). Although the method seems to be well-established, some choices or settings need to be justified or discussed. For example, it is clear how did the authors select the parcellation scheme. The authors used 200 ROIs in the present study, but used 360 ROIs in the previous one (Wang et al., 2021, PNAS). Also, Schaefer atlas offers the parcellation scheme from 100 to 1000 ROIs. Therefore, what was the consideration behind such choice? Would the authors assume the results (segregation vs. integration) are unaffected by number of ROIs (e.g., 200 vs. 400 vs. 1000)? Any evidence to support this?

2. A previous study identified elevated global brain signal variability in schizophrenia (Yang et al., 2014, PNAS, DOI: 10.1073/pnas.1405289111). I wonder if there is any connection between segregation and global brain signal variability.

3. A schematic illustration of the key results at the end of the Results section might be helpful.

Reviewer #2 (Remarks to the Author):

-- summary --

The presented manuscript reports a study that aims at characterizing the brain functional and genetic underpinnings of positive and negative symptoms in schizophrenia. To this aim, the authors used a nested-spectral partition approach to study dynamic transitions of functional segregation and integration in resting state fMRI. The results of this study show promising leads for better understanding the differences in the neural basis of positive and negative symptoms. Furthermore, the manuscript is well structured and written. Overall, I'm thus in favor of seeing this article in published form. However, there are some comments and critique that should be addressed prior to further consideration.

-- major comments --

My biggest concern is the very technical presentation and nature of the manuscript. While the advanced methodology is certainly a quite positive aspect of this study, the manuscript itself feels very much aimed at a methodologically specialized community, which makes it hard to follow for researchers without strong familiarity of the abbreviations or those who are missing intuition for the chosen methodological approach. I strongly recommend to strengthen the understandability of the methods to make the manuscript more approachable for Researchers generally interested in Schizophrenia.

-- minor/detailed comments --

generell

Keywords are missing.

Explanations for figures are rather sparse.

abstract

"Attention systems were associated with hallucinations, and default-mode network (DMN) and control systems were related to avolition."

 what metric of the networks was associated with these phenomena?

introduction

"... consistent and opposite results have been widely reported"

 sounds a bit contradictory. Please elaborate.

"However, whether positive and negative symptoms have opposite effects on segregation and integration remains unknown. Recently, a nested-spectral partition (NSP) method based on eigenmodes was proposed to detect hierarchical modules in brain networks and describe segregation and integration across multiple levels"

 While I believe it to be a very interesting idea to use NSP for this task, the sentence left me wondering if there are studies looking at this question with more more established method. It might make sense to directly compare graph theory results with your NSP-based analysis as a supplementary analyses. Please elaborate and consider adding this information.

methods

"AFNI (<http://afni.nimh.nih.gov/afni/>) and FSL (<http://www.fmrib.ox.ac.uk/fsl/>) were used to preprocess the resting-state fMRI data"

 Please report the entire preprocessing pipeline in detail, to make your study replicable.

"In each iteration of LOO-CV, one sample was selected as the test set,..."

 Please clarify what is meant with sample in this case. I believe this means one participant?

formula: "H ~ SANS + SAPS + SANS x SAPS + sex + age + FD"

 I might have overlooked it but in case: please define FD.

Abbreviations, Formula, Metrics.

 At times I'm having a hard time following what metrics were computed and what exactly they represent in terms of network behavior. In the current form, the paper is quite hard to follow in terms of the included graph or non-graph theoretical measures (for instance see result section)

results

NSP results.

 As I'm unfamiliar with this method, i would have expected some form of description what has actually come out of the nested-spectral partition in similar form as with other dimensionality reduction techniques.

figure 1. It is partially unclear to me what exactly is depicted in the subplots. Please consider adding some sentences about the metrics shown here (for example figure b - c).

discussion

"Here, we provided the first direct evidence that positive and negative symptoms are opposite on the brain while excluding their interaction, and DMN, negatively correlated with other systems may underlie the positive symptoms and negative symptoms, highly consistent with Ref77."

 This sentence feels quite convoluted. Furthermore, I'm uncertain what is meant with "..symptoms are opposite on the brain...". Please consider rephrasing and elaborating on what observation/finding is meant here exactly.

"This result greatly extends the understanding of schizophrenia and is helpful for developing treatment methods for both positive and negative symptoms wherein negative symptoms often persist after treatment with antipsychotic medication 80."

 This appears to imply that antipsychotic medication does not target the appropriate systems/neural agents associated with the negative symptom. I believe there is some research on this topic which might be beneficial to include and discuss at this point. For instance, please see: <https://www.nature.com/articles/s41598-019-51023-0>; <https://doi.org/10.1159/000440979>

Reviewer #3 (Remarks to the Author):

Summary:

This study compares a sample of schizophrenia patients with a matched set of healthy controls on measures of resting-brain connectivity using a nested-spectral partition approach (NSP), which permitted the characterization of hierarchical patterns of functional connectivity. The authors further compared these measures with positive and negative symptom scores and gene ontology analyses. The study finds differences in integrating and segregating processes in the brain patterns of healthy controls versus patients, and predicted symptom scores. Gene ontology indicated correlations between negative and positive symptoms mapped onto different cellular processes and phenotypes. The most striking finding was the ability to distinguish between the effects of negative and positive symptom scores on segregation and integration variability. The conclusions are well-couched in the literature and appropriate for the data and analyses provided.

Novelty/relevance:

NSP appears to be a relatively newly developed technique that shows promise for neuroimaging analyses and seems to have an advantage for characterizing the dynamics of segregation and integration in functional brain processes, rather than simply the amount of time in different network states. Given the heterogeneity of symptoms in schizophrenia, and the evidence for differences in network-level function, it is an interesting and appropriate approach. Combining this analysis with gene ontology is also innovative.

Other: The paper is extremely well-written, and the figures are clear and easy to comprehend even for workers not specifically in neuroimaging. I have no suggestions for changes; the manuscript is excellent.

Reviewer #1 (Remarks to the Author):

Wang et al. studied the brain's dynamic functional segregation and integration in schizophrenia patients, comparing to healthy controls. The authors utilized a nested-spectral partition (NSP) approach to detect hierarchical modules in an open resource fMRI dataset shared by UCLA. It was found that the brain networks of schizophrenic patients had a reduced global integration, accompanied by a more stable dynamic integrating process and a more variable segregating process. Brain network and gene ontology analyses were performed with a package of results. Linking brain networks with distinct positive and negative symptoms in schizophrenia appears to be novel. Overall, the manuscript is well-written. The analytical approach and findings are solid.

Response: Thank you for your positive evaluation of our work and for providing constructive comments. We have considered and addressed all of them carefully. Please see our following point-by-point responses.

I have a few methodological concerns to share.

1. The measures about hierarchical segregation and integration are mostly derived from the authors' previous study (Wang et al., 2021, PNAS). Although the method seems to be well-established, some choices or settings need to be justified or discussed. For example, it is clear how did the authors select the parcellation scheme. The authors used 200 ROIs in the present study, but used 360 ROIs in the previous one (Wang et al., 2021, PNAS). Also, Schaefer atlas offers the parcellation scheme from 100 to 1000 ROIs. Therefore, what was the consideration behind such choice? Would the authors assume the results (segregation vs. integration) are unaffected by number of ROIs (e.g., 200 vs. 400 vs. 1000)? Any evidence to support this?

Response: Thank you for this comment. First, we used the Schaefer200 atlas without any specific assumption. To indicate that our results in this manuscript are robust for different parcellation resolutions, we further calculated the results using the Schaefer500 atlas [1]. All results are similar for the parcellations of 200 and 500 regions (see Figs. R1-R5). Now, in the main text, we reported the results for 200 regions and added results for 500 regions to the Supplementary Materials (Figs. S1-

S5). In the main text, we have cited them in the caption of each figure: *see similar results for the brain parcellation of 500 regions in Fig. S, and in the Methods: and the results are similar for the brain parcellation of 500 regions (see Figs. S1-S5).*

Second, for the same fMRI data, finer parcellation of the brain would generate more modules with smaller sizes in high-order levels of brain functional networks, reflected by a larger segregation component and lower integration component (see Fig. R6). Therefore, we did not assume that the results (segregation vs. integration) are unaffected by the number of ROIs because they are actually dependent on the ROI number. However, our results for schizophrenia are robust for different brain ROI numbers. We have added this discussion to the Methods: *Notably, finer parcellation of the brain (i.e., 500 regions) would generate more modules with smaller sizes in higher-order levels of brain functional networks, accompanied by a larger segregation component and lower integration component (see Fig. S7). However, the results for schizophrenia are robust for different brain parcellations (see Figs. S1-S5).*

Reference:

1. Schaefer A, *et al.* Local-global parcellation of the human cerebral cortex from intrinsic functional connectivity MRI. *Cereb Cortex* **28**, 3095-3114 (2018).

Fig. R1. Dynamic segregation and integration for 500 brain regions. a) Comparisons of

integration strength H_{in} and **b**) segregation strength H_{se} between the healthy control (HC) and schizophrenia (SCH) groups. $**p<0.01$, $***p<0.001$. **c**) Relative changes in segregation/integration strengths from HC to SCH in seven functional systems. **d**) Comparisons of integration variability F_{in} and **e**) segregation variability F_{se} between the HC and SCH groups. $***p<0.001$. **f**) Relative changes in segregation/integration variabilities from HC to SCH in seven functional systems. All systems had significant alterations ($p<0.05$, FDR corrected). VIS-visual; MOT-motor; DOR-dorsal attention; SAL-salient attention; LIM-limbic; CON-control; DMN-default-mode network.

Fig. R2. Relationships between schizophrenia symptoms and brain networks with 500 regions. Correlations between hallucinations score and **a**) integration strength H_{in} , **b**) segregation strength H_{se} , **c**) integration variability F_{in} for the whole-brain (ALL) and seven functional systems. The red dots indicate significant correlations. **d**) Correlations of avolition score and **e**) alogia score to segregation variability F_{se} for the whole-brain (ALL) and seven functional systems.

Fig. R3. Predicting SANS/SAPS scores in the brain network with 500 regions. **a)** Four brain network measures integration strength H_{In} , segregation strength H_{Se} , integration variability F_{In} and segregation variability F_{Se} were used to predict the SAPS (upper panel) and SANS (lower panel) scores. The bar charts are correlations between SAPS/SANS scores and predicted scores using different brain measures. $**p<0.01$, $*p<0.05$. **b)** The correlation between SAPS (upper panel)/SANS (lower panel) and predicted scores for segregation variability F_{Se} . **c)** The weights of regions in the best predictions of SAPS (upper panel)/SANS (lower panel) scores were mapped to the brain surface. **d)** The summed weight of the regions in seven functional systems in the best predictions of SAPS (upper panel)/SANS (lower panel) scores.

Fig. R4. Effects of schizophrenia symptoms on brain networks with 500 regions. Effects of SANS/SAPS on **a)** dynamic integration variability F_{In} and **b)** segregation variability F_{Se} . **c)**

Coeffects of SANS/SAPS on integration variability F_{In} and segregation variability F_{Se} . The coefficients were obtained from **a)** and **b)** using PCA. The averaged coefficient of SANS and SAPS on regions within each system was marked. **d)** Interaction effect of SANS and SAPS on integration variability F_{In} and segregation variability F_{Se} .

Fig. R5. GO enrichment results for biological processes and human phenotypes using ToppGene. The top five biological processes and human phenotypes are similar to those for 200 regions.

Fig. R6. Higher segregation in larger brain functional networks. **a)** Comparisons of integration strength H_{In} and **b)** segregation strength H_{Se} between 200 and 500 regions in the HC group. * $p < 0.05$, *** $p < 0.001$.

2. A previous study identified elevated global brain signal variability in schizophrenia (Yang et al., 2014, PNAS, DOI: 10.1073/pnas.1405289111). I wonder if there is any connection between segregation and global brain signal variability.

Response: Thank you for this interesting comment. According to your suggestion,

we obtained the global signal time series from the preprocessed fMRI data and calculated the corresponding standard variance to represent the global signal fluctuation (GSF). Higher GSF significantly corresponds to stronger network integration and larger variability of the integration process (Table R1), supporting your comment that GSF is connected to network segregation and integration. In addition, GSF may contain clinically meaningful information, and regressing GSF does not necessarily always change between-group inferences [1]. Therefore, we did not remove the global signal. We have addressed this in the Methods: *Since the global whole-brain signal was related to brain network integration and segregation (Table S1) and may contain the clinical information of schizophrenia symptoms*^{37, 38, 39}, it was not removed from our analysis. We have added Table. R1 to the Supplementary Materials (Table S1).

Reference:

1. Yang GJ, *et al.* Altered global brain signal in schizophrenia. *Proceedings of the National Academy of Sciences* **111**, 7438-7443 (2014).

Table R1. Correlations between GSF and network measures: integration strength H_{in} , segregation strength H_{se} , integration variability F_{in} and segregation variability F_{se} in the whole-brain (ALL) and systems. Significant correlations are marked in red.

corr (p value)	ALL	VIS	MOT	DOR	SAL	LIM	CON	DMN
H_{in}	0.689 (0.001)	0.485 (0.001)	0.430 (0.002)	0.512 (0.001)	0.469 (0.001)	0.447 (0.001)	0.539 (0.001)	0.543 (0.001)
H_{se}	-0.593 (0.001)	-0.558 (0.001)	-0.441 (0.001)	-0.533 (0.001)	-0.502 (0.001)	-0.539 (0.001)	-0.582 (0.001)	-0.556 (0.001)
F_{in}	0.526 (0.001)	0.391 (0.005)	0.247 (0.084)	0.309 (0.029)	0.265 (0.063)	0.408 (0.003)	0.349 (0.013)	0.355 (0.011)
F_{se}	-0.522 (0.001)	-0.317 (0.025)	-0.404 (0.004)	-0.380 (0.006)	-0.300 (0.034)	-0.071 (0.625)	-0.234 (0.102)	-0.091 (0.530)

3. A schematic illustration of the key results at the end of the Results section might be helpful.

Response: Thank you for this comment. We have added the key points at the end of each section of the Results.

- 1) Analysis of dynamic segregation and integration:

Overall, schizophrenia, characterized by the loss of effective global integration, is related to a more stable global integration process and a more variable

segregation process, and the limbic system is most significantly changed. These alterations can be effectively detected by the NSP method relative to graph theory analysis.

2) Associations between dynamic brain networks and clinical symptoms in schizophrenia:

Overall, brain dynamic networks are related to key positive symptoms (i.e., hallucinations) and key negative symptoms (i.e., avolition). More serious hallucinations correspond to higher integration strength in attention systems. Weaker avolition is associated with higher dynamic variability of the segregation process in the default-mode network (DMN) and control systems.

3) Predictions of positive and negative symptoms:

Overall, compared to graph theory measures, NSP-based features can better predict the SAPS and SANS scores, suggesting potential biomarkers for schizophrenia. The DMN, visual and limbic systems significantly participate in the predictions, reflecting their important roles in schizophrenia.

4) Opposite effects of positive and negative symptoms on the brain:

Overall, positive and negative symptoms of schizophrenia have opposite effects on the functional organization of resting-state brains. SAPS predominantly affects the motor system; SANS dominantly affects the control system; SANS×SAPS interaction interactively affects both attention systems.

5) Linking effects of schizophrenia symptoms to gene transcriptional profiles:

Overall, the effect of SANS is related to synaptic signaling processes and abnormal autistic, aggressive and violent behaviors; the effect of SAPS is related to metabolic precursors, hyperammonemia and acidosis; and the SANS×SAPS interaction effect is related to cellular localization and transport and abnormal motor function. All these phenotypes have been found in schizophrenia ^{48, 49, 50, 51,52}.

Reviewer #2 (Remarks to the Author):

The presented manuscript reports a study that aims at characterizing the brain functional

and genetic underpinnings of positive and negative symptoms in schizophrenia. To this aim, the authors used a nested-spectral partition approach to study dynamic transitions of functional segregation and integration in resting state fMRI. The results of this study show promising leads for better understanding the differences in the neural basis of positive and negative symptoms. Furthermore, the manuscript is well structured and written. Overall, I'm thus in favor of seeing this article in published form. However, there are some comments and critique that should be addressed prior to further consideration.

Response: Thank you for the positive evaluation of our work. We have carefully addressed all your comments. Please see our following point-by-point responses.

1. My biggest concern is the very technical presentation and nature of the manuscript. While the advanced methodology is certainly a quite positive aspect of this study, the manuscript itself feels very much aimed at a methodologically specialized community, which makes it hard to follow for researchers without strong familiarity of the abbreviations or those who are missing intuition for the chosen methodological approach. I strongly recommend to strengthen the understandability of the methods to make the manuscript more approachable for Researchers generally interested in Schizophrenia.

Response: Thank you for this comment. To improve the readability of our method, 1) we split the “Hierarchical segregation and integration components” section into two shorter parts and reorganized the description, e.g., *Therefore, for a single FC network, we obtained the separated integration component H_m and segregation component H_{se}* ; 2) in the Results section, we added more descriptions about the NSP method: *The NSP method can extract the separated segregation and integration components from a single brain FC network without any threshold, which is able to study the network dynamic transitions in two separate dimensions*; 3) we described the measure names and abbreviations together, e.g., integration strength H_m , integration variability F_m . Please see our revisions in the main text

marked in blue.

2. Keywords are missing.

Response: We have added keywords: *schizophrenia, MRI, nested-spectral partition, functional connectivity, gene ontology.*

3. Explanations for figures are rather sparse.

Response: Thank you for this comment. We have expanded the captions of Figs. 1-5; please see our revisions in the main text marked in blue.

4. "Attention systems were associated with hallucinations, and default-mode network (DMN) and control systems were related to avolition." what metric of the networks was associated with these phenomena?

Response: Thank you for this comment. We have changed this description as follows: *Hallucinations were associated with higher integration in attention systems, and avolition was related to a more variable segregating process in default-mode network (DMN) and control systems.*

5. "... consistent and opposite results have been widely reported" sounds a bit contradictory. Please elaborate.

Response: We have changed this description as follows: *but opposite results have been widely reported.*

6. "However, whether positive and negative symptoms have opposite effects on segregation and integration remains unknown. Recently, a nested-spectral partition (NSP) method based on eigenmodes was proposed to detect hierarchical modules in brain networks and describe segregation and integration across multiple levels" While I believe it to be a very interesting idea to use NSP for this task, the sentence left me wondering if there are studies looking at this question with more more established method. It might make sense to directly compare graph theory results with your NSP-based analysis as a supplementary analysis. Please elaborate and consider adding this information.

Response: Thank you for this advanced comment. We have compared the graph theory measures (participation coefficient and degree) with our NSP measures. Consistently, we defined degree strength as H_{Deg} , participation coefficient strength as H_{PC} , degree variability as F_{Deg} and participation coefficient variability as F_{PC} . The comparisons are as follows:

- 1) There were no significant differences in graph theory measures between the healthy control (HC) and schizophrenia (SCH) groups, but our NSP method detected the differences (Fig. R1).
- 2) Graph theory and NSP measures have consistent correlations to key positive symptoms (*i.e.*, hallucinations); graph theory measures are also related to positive symptoms thought disorder and blunted affect, but only NSP measures are related to negative symptoms (*i.e.*, avolition, Fig. R2).
- 3) In the machine learning model, the NSP measures outperformed the classical graph theory measures in predicting the SAPS and SANS scores (Fig. R3).
- 4) In the multiple regression model, opposite effects of SAPS and SANS on regions were found for both graph theory and NSP measures (Fig. R4). However, for graph theory measures, the number of significantly related genes is not enough to statistically enrich biological processes and human phenotypes in ToppGene. For example, we identified 1780 SANS-related genes for our NSP measures but only 194 SANS-related genes for the graph theory measures. Therefore, we did not provide the GO results for graph theory measures.

We have added the above figures to the Supplementary Materials (see Figs. S9 and S11-S13) and added the above descriptions to the captions of the figures. In the main text, we have cited these results:

- 1) Analysis of dynamic segregation and integration:

We also calculated the corresponding graph theory measures: degree, participant coefficient, variability of degree and variability of participant coefficient, and found that there were no significant differences in graph theory

measures between the HC and SCH groups (see Fig. S9).

2) Associations between dynamic brain networks and clinical symptoms in schizophrenia:

Graph theory measures have consistent correlations to the key positive symptoms (i.e., hallucinations) but are not related to negative symptoms (i.e., avolition, see Fig. S11).

3) Predictions of positive and negative symptoms:

Our results showed that segregation variability F_{se} has the best prediction of the SAPS score ($r = 0.654, p < 0.001$) and SANS score ($r = 0.659, p < 0.001$) (Fig. 3a, b) and outperforms the classical graph theory measures (see Fig. S12).

4) Opposite effects of positive and negative symptoms on the brain:

These opposite effects of SAPS and SANS on regions were also found in graph theory measures (see Fig. S13).

5) Linking effects of schizophrenia symptoms to gene transcriptional profiles:

For graph theory measures, the number of significantly related genes is not enough to statistically enrich biological processes and human phenotypes, e.g., only 194 SANS-related genes were identified for the graph theory measures.

Fig. R1. Brain network measures for graph theory. a) Comparisons of NSP measures and b) graph theory measures between HC and SCH groups. ** $p < 0.01$ *** $p < 0.001$.

Fig. R2. Associations between resting-state brain networks and schizophrenia symptoms.

a) Correlations of symptom scores to NSP measures and **b)** graph theory measures for the whole-brain (ALL) and seven functional systems.

Fig. R3. Prediction of SANS and SAPS scores. **a)** Eight brain network measures, integration strength H_{In} , segregation strength H_{Se} , integration variability F_{In} , segregation variability F_{Se} , degree strength H_{Deg} , participation coefficient strength H_{PC} , degree variability F_{Deg} and participation coefficient variability F_{PC} , were used to predict the SAPS and **b)** SANS scores. The bar charts are correlations between real and predicted scores. *** $p < 0.001$, ** $p < 0.01$, * $p < 0.05$.

Fig. R4 Effects of schizophrenia symptoms on brain networks. The correlation between the effect of SANS and the effect of SAPS for a) NSP and b) graph theory measures.

7. “AFNI (<http://afni.nimh.nih.gov/afni/>) and FSL (<http://www.fmrib.ox.ac.uk/fsl/>) were used to preprocess the resting-state fMRI data”
 → Please report the entire preprocessing pipeline in detail, to make your study replicable.

Response: Thank you for your suggestion. We have added more descriptions to the data preprocessing process: Resting-state fMRI data were processed using FSL (<http://www.fmrib.ox.ac.uk/fsl/>) and AFNI (<http://afni.nimh.nih.gov/afni/>) software in the Ubuntu 14.04 system²⁹. The procedure included 1) slice-timing correction to the median slice; 2) motion correction; 3) segmenting the anatomical image; 4) Montreal Neurological Institute (MNI) normalization; 5) spatial smoothing using a Gaussian kernel with a 6-mm full width at half maximum (FWHM); 6) bandpass filtering (0.01-0.1 Hz); and 7) elimination of 6 rigid body motion correction parameters and the signal from the white matter and a ventricular region of interest using linear regression. The mean framewise displacement (FD) was 0.160 ± 0.159 mm for the healthy control group and 0.267 ± 0.215 mm for the schizophrenia group. The difference in FD between the two groups was significant (two-sample t test, $t(98) = 2.779$, $p = 0.004$). Thus, an analysis of covariance (ANCOVA) was carried out for the group comparison. Since the global whole-brain signal was related to brain network measures (Table S1) and may contain the clinical information of schizophrenia symptoms^{37,38,39}, it was not removed from our analysis.

8. "In each iteration of LOO-CV, one sample was selected as the test set,... " Please clarify what is meant with sample in this case. I believe this means one participant?

Response: Thank you for this comment. The samples indicate the participants. We have changed “sample” to “participant” in the Methods: *one participant was selected as the test set, and the remaining participants were selected as the training set.*

9. formula: " $H \sim \text{SANS} + \text{SAPS} + \text{SANS} \times \text{SAPS} + \text{sex} + \text{age} + \text{FD}$ " I might have overlooked it but in case: please define FD.

Response: We apologize for this description. FD indicates the mean framewise displacement. We have added more details of FD in this formula: *FD is the mean framewise displacement.*

10. Abbreviations, Formula, Metrics.  At times I'm having a hard time following what metrics were computed and what exactly they represent in terms of network behavior. In the current form, the paper is quite hard to follow in terms of the included graph or non-graph theoretical measures (for instance see result section)

Response: We apologize for those descriptions. We now report the network measure names and abbreviations together, e.g., integration component H_m and integration variability F_m (also see our response to question 1). All revisions have been marked in blue in the manuscript.

11. NSP results.  As I'm unfamiliar with this method, i would have expected some form of description what has actually come out of the nested-spectral partition in similar form as with other dimensionality reduction techniques.

Response: In the Results section, we have added a general description about NSP output: *The NSP method can extract the separated segregation and integration components from a single brain FC network without any threshold, which is able to study the network dynamic transitions in two separate dimensions (also see our response to question 1).*

12. figure 1. It is partially unclear to me what exactly is depicted in the subplots. Please consider adding some sentences about the metrics shown here (for example figure b - c).

Response: We apologize for the insufficient descriptions in the figure captions. Fig. 1b shows the comparisons of integration strength H_{in} between healthy controls and schizophrenia patients. In this figure, each point indicates the integration strength H_{in} of an individual, and the box plot for each group is also shown. Figs. 1c-1f show similar results for segregation strength H_{se} , integration variability F_{in} and segregation variability F_{se} . We have added these descriptions to the caption of Fig. 1b-1c: **b) Comparisons of integration strength H_{in} , c) segregation strength H_{se} between HC and SCH groups. Each point indicates the value of an individual, and the boxplot of each group was also provided. ** indicates a significant group difference with $p < 0.01$, *** $p < 0.001$. Similarly, in the main text, we have carefully revised all figure captions (also see our response to question 3).**

13. "Here, we provided the first direct evidence that positive and negative symptoms are opposite on the brain while excluding their interaction, and DMN, negatively correlated with other systems may underlie the positive symptoms and negative symptoms, highly consistent with Ref77."

 This sentence feels quite convoluted. Furthermore, I'm uncertain what is meant with "...symptoms are opposite on the brain...". Please consider rephrasing and elaborating on what observation/finding is meant here exactly.

Response: We are grateful for the comment. We have revised the description: *Here, we provided the first direct evidence that positive and negative symptoms of schizophrenia have opposite effects on the functional organization of resting-state brains while excluding their interaction, especially that the DMN has opposite contributions to the predictions of SANS and SAPS scores. The DMN is negatively correlated with other systems⁸², and its abnormality may underlie the positive*

symptoms^{71, 73, 83} and *negative symptoms*^{76, 77, 78} of schizophrenia. Hare SM et al. reported that FC with a 4 s lag between the anterior DMN and posterior DMN was negatively associated with the severity of disordered thought and attentional deficits, and FC with a 2 s lag between the anterior DMN and salience network was positively related to the severity of flat affect and bizarre behavior⁷⁹, which is highly consistent with our observations of opposite functions of the DMN on positive and negative symptoms.

14. "This result greatly extends the understanding of schizophrenia and is helpful for developing treatment methods for both positive and negative symptoms wherein negative symptoms often persist after treatment with antipsychotic medication⁸⁰."
 This appears to imply that antipsychotic medication does not target the appropriate systems/neural agents associated with the negative symptom. I believe there is some research on this topic which might be beneficial to include and discuss at this point. For instance, please see:

<https://www.nature.com/articles/s41598-019-51023-0>;

<https://doi.org/10.1159/000440979>

Response: Thank you for providing these relevant references. Vanes L.D. et al. [1] studied fMRI activation on a standard Stroop task and its relationship with positive and negative symptoms in early psychosis and chronic schizophrenia patients, but they found a nonsignificant relationship between brain activations and negative symptoms. Shaffer J.J. et al [2] studied the relationship between individual negative symptoms and functional brain activity during an auditory oddball task and found a negative correlation between negative symptoms (anhedonia/asociality) and activity in the posterior cingulate and precuneus, part of the DMN. We have cited these two papers in the main text: *In particular, negative symptoms often persist after treatment with antipsychotic medication*⁸⁴. *Even though negative symptoms (e.g., anhedonia/asociality) were found to be related to the posterior cingulate and precuneus, part of the DMN, in a two-tone auditory oddball task, identifying reliable targets of regions for treatment remains a challenge in the clinic*⁸⁵. Our

results greatly extend the understanding of schizophrenia and provide that distinct DMN regions may be targets for positive and negative symptoms.

References:

1. Reckless GE, Andreassen OA, Server A, Østefjells T, Jensen J. Negative symptoms in schizophrenia are associated with aberrant striato-cortical connectivity in a rewarded perceptual decision-making task. *NeuroImage: Clinical* **8**, 290-297 (2015).
2. Shaffer JJ, *et al.* Neural Correlates of Schizophrenia Negative Symptoms: Distinct Subtypes Impact Dissociable Brain Circuits. *Molecular Neuropsychiatry* **1**, 191-200 (2015).

Reviewer #3 (Remarks to the Author):

This study compares a sample of schizophrenia patients with a matched set of healthy controls on measures of resting-brain connectivity using a nested-spectral partition approach (NSP), which permitted the characterization of hierarchical patterns of functional connectivity. The authors further compared these measures with positive and negative symptom scores and gene ontology analyses. The study finds differences in integrating and segregating processes in the brain patterns of healthy controls versus patients, and predicted symptom scores. Gene ontology indicated correlations between negative and positive symptoms mapped onto different cellular processes and phenotypes. The most striking finding was the ability to distinguish between the effects of negative and positive symptom scores on segregation and integration variability. The conclusions are well-couched in the literature and appropriate for the data and analyses provided.

Response: Thank you for your high evaluation of our work.

Novelty/relevance:

NSP appears to be a relatively newly developed technique that shows promise for neuroimaging analyses and seems to have an advantage for characterizing the dynamics of segregation and integration in functional brain processes, rather than simply the amount of time in different network states. Given the heterogeneity of symptoms in schizophrenia and the evidence for differences in network-level function, it is an

interesting and appropriate approach. Combining this analysis with gene ontology is also innovative.

Other: The paper is extremely well-written, and the figures are clear and easy to comprehend even for workers not specifically in neuroimaging. I have no suggestions for changes; the manuscript is excellent.

Response: Thank you for your comment.

REVIEWERS' COMMENTS:

Reviewer #1 (Remarks to the Author):

The authors have done a great job in addressing my concerns.
Zirui Huang

Reviewer #3 (Remarks to the Author):

I was very impressed with the initial version of the manuscript and did not have any revision recommendations. I see in this iteration the authors have addressed the concerns other reviewers had about readability and assessing/comparing alternative parcellations; once again I am happy to recommend this article for publication.